# A non-destructive image-based approach to quantify blood meal size in *Lutzomyia longipalpis* (Diptera: Psychodidae)

Lidiane Medeiros da Costa[1], Maurício Roberto Viana Sant'Anna[1/+], Laura Aquino Generoso[2], Geisler Peixoto da Cruz[1], Grasielle Caldas D'Ávila Pessoa[3], Ricardo Nascimento Araujo[4], Nelder de Figueiredo Gontijo[1], Marcos Horacio Pereira[1/+]

[1]Universidade Federal de Minas Gerais, Instituto de Ciências Biológicas, Departamento de Parasitologia, Laboratório de Fisiologia de Insetos Hematófagos, Belo Horizonte, MG, Brasil
[2]Pontifícia Universidade Católica de Minas Gerais, Belo Horizonte, MG, Brasil
[3]Universidade Federal de Minas Gerais, Instituto de Ciências Biológicas, Departamento de Parasitologia, Laboratório de Entomologia Médica, Belo Horizonte, MG, Brasil
[4]Universidade Federal de Minas Gerais, Instituto de Ciências Biológicas, Departamento de Parasitologia, Laboratório de Artrópodes Hematófagos, Belo Horizonte, MG, Brasil

**BACKGROUND** Phlebotomine sand flies are hematophagous vectors of major human pathogens, including *Leishmania* spp., with blood ingestion essential for reproduction and vector competence. Accurate quantification of blood meal volume is crucial for understanding physiological processes and transmission dynamics.

**OBJECTIVES** Here, we introduce a novel, non-destructive image-based method to estimate blood intake in *Lutzomyia longipalpis*, the principal vector of *Leishmania infantum* in the Americas.

**METHODS** High-resolution images of unfed and blood-fed females were analysed using Fiji ImageJ (open-source software) when several morphometric parameters were measured and validated against biochemical haemoglobin (Hb) quantification.

**FINDINGS** Blood-fed females exhibited a 56.6% increase in abdominal width and a shift toward a rounded body shape, which was strongly correlated with a visible transilluminated abdominal area ($R^2 = 0.92$). Some parameters, such as mean grey value and abdominal length, showed a low to moderate correlation with Hb content ($R < 0.60$). However, the correlation with abdominal area and width was $R \approx 0.90$, indicating those are reliable parameters that can be used to estimate blood intake by *Lu. longipalpis* females.

**MAIN CONCLUSIONS** Unlike spectrophotometric methods, this approach preserves specimen integrity, which, in theory, enables longitudinal studies on physiology and host-parasite interactions. This methodology offers a reliable, scalable, and cost-effective alternative for estimating blood meal.

Key words: *Lutzomyia longipalpis* - blood meal size - image analysis - haemoglobin quantification

Phlebotomine sand flies are haematophagous dipterans belonging to the Family Psychodidae. More than 900 species have been identified, with a global distribution that excludes only Antarctica. The greatest abundance and diversity occur in tropical and subtropical regions.[1]

Certain phlebotomine species act as vectors of pathogens responsible for viral (*Phlebovirus* spp.), bacterial (*Bartonella* spp.), and protozoan (*Leishmania* spp.) infections in both humans and animals.[2,3,4,5] Leishmaniasis remains an important global health issue, with over 272,000 new human cases reported in 2023.[6] *Lutzomyia longipalpis* is recognised as the principal vector of *Leishmania infantum* in the Americas, the causative agent of American visceral leishmaniasis (AVL).[7] Notably, *Lu. longipalpis* exhibits a high degree of adaptation to urban environments, with multiple studies indicating a substantial expansion of its urban distribution since the 1980s, which has significantly altered the transmission dynamics and epidemiological patterns of visceral leishmaniasis.[8]

Moreover, increasing molecular and ecological evidence supports the hypothesis that *Lu. longipalpis* comprises a species complex of cryptic taxa with subtle morphological similarities, potentially accounting for variations in vectorial capacity across distinct geographic populations.[7,9,10,11]

In phlebotomine sand flies, adult males and females feed on carbohydrate-rich sources such as nectar, plant sap, and aphid secretions. Carbohydrates are initially directed to the crop in the anterior midgut, where they are stored and later transferred to the midgut for digestion.[12] Blood is obtained through telmophagy, where superficial dermal layers are disrupted using short, rig-

**doi:** 10.1590/0074-02760250158
**Financial support:** CAPES, FAPEMIG, CNPq, INCTEM.
**+ Corresponding authors:** marcoshp@icb.ufmg.br | https://orcid.org/0000-0003-4843-0088 / mrvsantanna@gmail.com | https://orcid.org/0000-0002-2504-0979

**Handling editor:** Adeilton Alves Brandão | https://orcid.org/0000-0001-5877-607X

id mouthparts to create blood pools that are ingested.[13] This feeding method facilitates ingesting *Leishmania* parasites in the skin.[14]

Under laboratory conditions, blood meals typically last 1-5 min, during which females may ingest an amount of blood approximately equivalent to their body weight (0.1-0.6 mg).[15] Blood is directed to the midgut, leading to abdominal distension mediated by the pleural membranes.[16] Blood digestion takes 48-72 h. Digestion products such as amino acids, fatty acids, and sugars are absorbed by enterocytes and transported to support ovarian maturation.[17,18,19,20]

The number of eggs produced is proportional to the volume of blood ingested, with a maximum determined by the number of ovarioles (approximately 50).[21,22,23] In *Lu. longipalpis*, females are anautogenous, exhibiting gonotrophic concordance, which means they lay eggs only after a blood meal during their first reproductive cycle.[24] Typically, 30-70 eggs are laid per gonotrophic cycle.

Beyond reproductive consequences, the volume of blood meal can influence the efficiency of pathogen transmission.[25] Therefore, quantifying blood intake is crucial for understanding vector biology, population dynamics, and disease epidemiology, as well as for developing effective control strategies.[26,27]

Several methods have been employed to estimate blood meal volume, including gravimetric analysis,[17-27] radioisotope labelling,[28] vital dyes,[29] and spectrophotometric haemoglobin (Hb) quantification.[27,30,31,32,33] For instance, Volfová[27] correlated Hb dosage with body weight gain in ten sand fly species fed on various hosts. Drabkin's solution, containing cyanide and ferricyanide, is widely used to form a stable coloured complex with Hb for quantification.[34,35,36] Martin-Martin[33] employed this reagent to estimate blood volumes in *Lu. longipalpis* in a study assessing the immunogenicity of salivary proteins and their impact on *L. major* infection.

While effective, many existing techniques for estimating blood meal size in sand flies are invasive, destructive, or chemically dependent, limiting their applicability in live-insect studies. In response, we propose an innovative, non-destructive, and chemical-free image-based method suitable for live and anesthetised female sand flies. This method, which involves capturing high-resolution images of females after blood feeding on anesthetised hamsters and analysing them using Fiji ImageJ (open-source software),[37] offers a promising alternative to existing techniques in insect research.

## MATERIALS AND METHODS

*Sand fly rearing* - *Lu. longipalpis* specimens originating from Teresina (Piauí, Brazil) were used in this study. Sand flies were maintained in a closed colony at the insectary of the Laboratory of Haematophagous Insect Physiology (ICB-UFMG) under controlled temperature conditions (25 ± 1ºC), according to the protocol described by Modi and Tesh.[38] Adults were fed ad libitum with a 15% sucrose solution on cotton pads and kept under a 12:12 h light/dark photoperiod. Blood feeding was performed weekly using hamsters (*Mesocricetus auratus*), aged four-eight weeks and weighing between 100 and 150 g. The animals were anaesthetised intraperitoneally with sodium thiopental (Thiopentax®, Cristália, Brazil), prepared at a concentration of 50 mg/mL and administered at a dose of 100 μL per 100 g of body weight (50 mg/kg). All procedures followed the guidelines established by the Animal Ethics Committee of the Federal University of Minas Gerais (CEUA protocol 262/2021).

*Feeding assay* - For experimental procedures, females aged five-seven days post-emergence were selected. Insects were divided into two groups (n = 60 females and 10 males per group) and housed in plastic containers (11 cm height, 8.5 cm diameter) covered with mesh fabric lids. The sugar solution was removed 12 h before the assay. In the control group (CG), females were maintained under fasting conditions, while in the experimental group (EG), females were allowed to feed on hamster for 1 h. The hamster used for blood feeding had no prior contact with sand flies or anaesthetic agents.

*Image acquisition and analysis* - To compare meal size estimation between image analysis and Hb quantification, insects from both groups were euthanised immediately after the feeding assays by freezing at -20ºC for 10 min. Females were subsequently positioned in lateral decubitus on a Petri dish containing graph paper as a size reference. The plate was positioned on a transilluminator with fixed bottom illumination, at a distance of 6.3 cm from the Casio Ex-FH20 camera. Images were captured at a resolution of 3456 × 2592 pixels with an exposure compensation of +2 exposure value (+2 EV). White balance and ISO settings were kept in automatic mode. Captured images were processed and analysed using Fiji ImageJ software (https://imagej.net/) (version 1.53f51). Image parameters related to size, shape, and colour were obtained by selecting the following features using the "Set Measurements" tool: area; shape descriptors (circularity and roundness); integrated density; mean grey value; perimeter and fitting an ellipse. To improve visualisation of small amounts of ingested blood in females, the pseudo flat-field correction tool was used, which allows background resulting from uneven illumination to be subtracted from both greyscale and true-colour images. The graphical representations of the data were produced using GraphPad Prism 10.6.1 software.

*Selection of regions of interest (ROIs)* - Following image calibration (histogram and scale adjustment), processing was performed (math + max 200 + brightness/contrast adjustments) to delineate ROIs. Image segmentation tools (thresholding or colour thresholding), which use grey-level intensity (8-bit images) or colour information from red, green and blue (RGB) images to separate the object of interest from the background, were applied. Three ROIs were selected [Supplementary data (Figure)]: the whole body excluding appendages (TBS); the abdominal region alone (ABS); the red-coloured abdominal region corresponding to the ingested blood volume visible through cuticle transparency (RCS). A standardised protocol was followed to define the ROIs, as illustrated in Fig. 1. For TBS: wings were removed (polygon selection + clear), a top-hat filter was applied to eliminate bristles

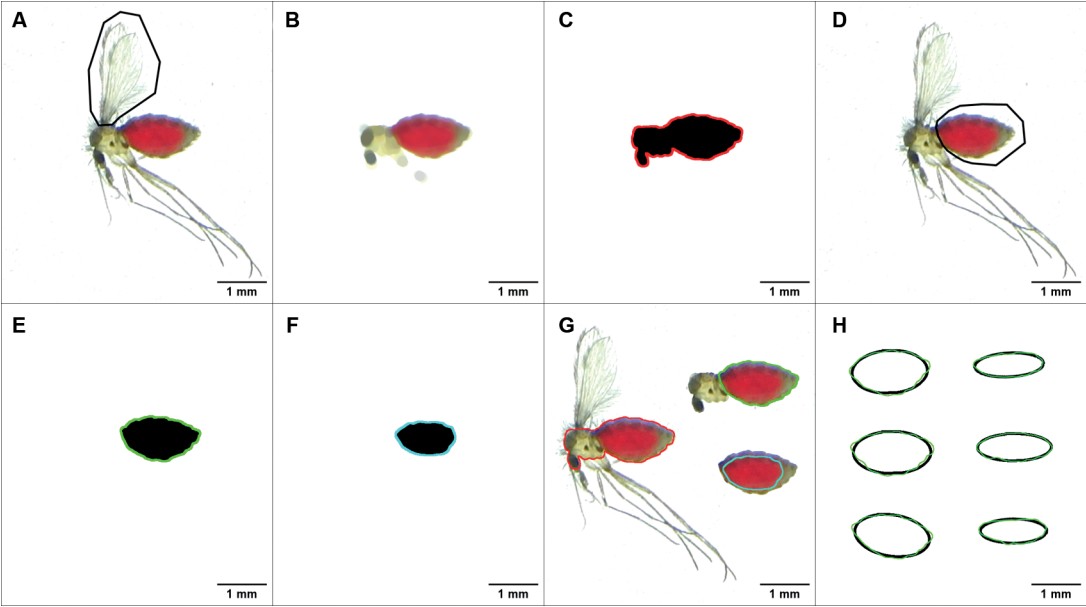

Fig. 1: steps for obtaining the regions of interest (ROIs) in the body of *Lutzomyia longipalpis*. (A-C) Selection of the contour of the entire body (TBS) excluding appendages; (D-E) selection of the abdominal contour (ABS); (F) selection of the contour of the red-coloured abdominal region (RCS) in blood-fed females; (G) localisation of the ROIs (TBS, ABS, and RCS) relative to the body parts/regions of a blood-engorged female sand fly; (H) comparison between the contours of the ABS ROIs (green lines) of blood-fed (left) and non-blood-fed (right) females with the corresponding ellipses generated by the "Fit Ellipse" tool (black lines). The Roman numerals I and II represent the major (~length) and minor (~width) axes.

and appendages, and the body contour was defined (conversion to 8-bit image + threshold + wand tracing tool) (Fig. 1A-C). For ABS: the abdomen was isolated from the thorax (polygon selection + clear outside), bristles were removed or attenuated (median filter), and the abdominal contour was delineated (8-bit conversion + threshold + wand tracing tool) (Fig. 1D-E). For RCS: the red-coloured area within ABS was selected using the colour threshold, clear outside and wand tracing tool (Fig. 1F). After selection, ROIs were compared with the original RGB images, corrected when necessary, and saved.

*Hb quantification* - To estimate the amount of blood ingested by the sand flies, Hb quantification was performed using Drabkin's reagent, following the manufacturer's instructions (Labtest Diagnóstica S.A). Each insect from the control and experimental groups was placed in a 2 mL Eppendorf tube containing 20 µL of Drabkin's reagent and labelled with the same identification number used in the captured images. The samples were then homogenised using a micro-homogeniser. After homogenisation, an additional 380 µL of Drabkin's reagent was added to each tube, followed by centrifugation at 10,000g for 5 min. Subsequently, 200 µL of the supernatant was transferred to a well of a 96-well microplate. The standard curve was generated by performing a two-fold serial dilution of heparinised human blood (0.2 µL/mL) across 12 wells of the microplate containing Drabkin's reagent (final volume of 200 µL), corresponding to a range of 5.0 to 0.002 µL of blood. After preparation, the optical density (OD) of the resulting solution was measured at 540 nm.

*Statistical analysis* - Data were analysed using GraphPad Prism 10.6.1 software. The Kolmogorov-Smirnov test assessed data normality. The t-test was used to test variables with a normal distribution. In cases where variables did not present a normal distribution, analyses were performed using the Mann-Whitney test. $P < 0.05$ was considered significant. The degree of association and the relationship between parameters obtained from image analysis and those from the Hb OD at 540 nm were assessed using Pearson correlation tests. Observations with absolute studentised residuals > 3 and a Cook's distances > 4/n were considered outliers,[39] and the linear regressions and $R^2$ values were re-estimated after excluding outliers from the dataset.

## RESULTS

*Image-based assessment of blood feeding in CG and EG females* - Lateral-view images of *Lu. longipalpis* females enabled clear visual differentiation between unfed and blood-fed individuals. Unfed females (CG group) exhibited variable abdominal profiles: some showed collapsed abdomens, where only tergal plates were discernible with poorly defined external plates and no visible pleural membrane (Fig. 2AI), while others displayed moderately expanded abdomens, characterised by a distinct separation between tergal and external plates via a visible pleural membrane spanning all abdominal segments (Fig. 2AII-AIII). In blood-fed females (EG), the most remarkable abdominal expansion consistently occurred between the fourth and fifth abdominal segments (Fig. 2B).

The use of transmitted light during image acquisition facilitated the detection of reddish coloration in the

thoracic and abdominal midgut (Fig. 2B), indicative of blood ingestion. Even females with minimal blood intake and no remarkable abdominal distension were reliably identified by applying image brightness and contrast adjustments (Fig. 2C). All females exposed to the host (EG) showed evidence of blood ingestion, as confirmed by image-based analysis.

*Blood feeding-induced changes in body size* - Quantitative analysis of image-derived parameters from TBS (whole body excluding appendages and ABS (abdomen) ROIs revealed substantial morphological differences between unfed (CG) and blood-fed (EG) females (Fig. 3).

Blood-fed females displayed a larger mean area for TBS (an increase of 37.8%) and ABS (an increase of 65.9%) compared to the controls (Fig. 1A-B). However, when the difference between TBS and ABS areas was analysed, values were similar (1.4%) (Fig. 3C), suggesting that the overall increase was primarily attributable to abdominal expansion following blood ingestion. Differences in abdominal dimensions were markedly greater in width (an increase of 56.6%) compared to length (an increase of 5.6%) between the EG and CG groups (Fig. 3D-E). As abdominal expansion increases the width-to-length ratio as a proxy for abdominal distension, blood-fed females exhibited significantly higher "circularity" ($0.71 \pm 0.04$ vs. $0.60 \pm 0.03$; Mann-Whitney test, $p < 0.01$) and "roundness" ($0.52 \pm 0.04$ vs. $0.35 \pm 0.03$; Mann-Whitney test, $p < 0.01$) compared to controls. Moreover, a distinct reddish abdominal coloration (mean grey value $93.4 \pm 11.00$ vs $142.1 \pm 10.36$; Mann-Whitney test, $p < 0.001$) was observed in engorged females, contrasting with the brownish abdomen of unfed specimens.

The abdominal expansion observed in fed females is related to blood intake; the mean OD value from the Hb measurements was 63.3% higher in the EG than in the CG (Fig. 3F).

*Analysis of blood-fed females based on abdominal colour* - To ensure consistency, one individual (the first from left to right in Fig. 2B) was identified as an outlier (absolute studentised residual > 3 and Cook's distance > 4/n in both OD × ABS and OD × RCS regressions) and was excluded from subsequent analyses. This specimen's OD value (0.15 at 540 nm) was much higher than expected given its very low ABS and RCS values (0.52 and 0.13 mm², respectively). This divergence may have been caused by pipetting or reading errors, leading to an overly high OD for this individual. Analysis of the remaining blood-fed females (EG; n = 59) showed that the red-coloured abdominal region (RCS; $0.70 \pm 0.20$ mm²) represented, on average, 79.54% of the total abdominal area (ABS; $0.86 \pm 0.15$ mm²), ranging from 45.30% to 92.98%. The RCS and ABS areas showed a strong positive correlation ($R^2 = 0.93$) (Fig. 4).

*Regression analysis of abdominal expansion parameters* - Since the abdomen's lateral profile of insects (CG and EG) fitted an elliptical shape, the "Fit Ellipse" tool was employed to estimate the length (major axis) and width (minor axis) of the sand flies' abdomens (Fig. 1H). Regression analysis indicated that ab-

dominal expansion in blood-fed females was strongly associated with an increase in width ($R^2 = 0.92$) rather than length ($R^2 = 0.54$) (Fig. 5), highlighting width as the major contributor to abdominal growth.

*Correlation between Hb quantification and image-derived parameters* - Table presents a Pearson correlation analysis between Hb OD values and image-derived parameters from abdominal regions (ABS and RCS) of sand flies. Chosen from the parameters available in the "Set Measurements" of Fiji Imagej menu, those that reach correlations greater than ~0.7 in ABS or RCS regions. Two of these parameters (Area and Width) showed very strong correlations (R = 0.87) between blood meal size and the abdominal areas (ABS or RCS). In addition, the linear regression that best fit the relationship between Hb OD values and imaging abdominal characteristics of blood-fed females was obtained using ABS and RCS Areas ($R^2 \sim 0.75$) (Fig. 6).

## DISCUSSION

Phlebotomine sand flies are principal vectors of several medically important pathogens, including phleboviruses, bacteria, and protozoans such as *Leishmania*. Blood feeding is one key determinant of their reproductive success and vectorial capacity. This physiological process initiates a cascade of events relevant to host-parasite interactions and pathogen transmission. This study introduces and validates a novel image-based methodology to assess morphological changes induced by blood ingestion in *Lu. longipalpis* females, benchmarking its accuracy against standard biochemical Hb quantification.

Our findings reveal that blood-feeding elicits substantial and quantifiable morphological alterations, particularly in abdominal width. Engorged females exhibited a 56.6% increase in abdominal width, while abdominal length increased by only 5.6%. This asymmetrical expansion led to a pronounced change in body conformation, shifting towards a more rounded morphology, which was robustly captured through circularity and roundness metrics derived from lateral-view images.

A markedly strong correlation ($R^2 = 0.93$) was observed between abdominal distension and the area of transilluminated reddish coloration, indicating that image-based analysis can serve as a reliable and non-invasive proxy for estimating blood meal volume. Remarkably, this approach enables the detection of blood-fed individuals even without visually obvious abdominal enlargement, addressing a critical limitation in conventional assessment methods that rely on gross visual inspection or gravimetric measurements.

Further supporting the validity of this approach, we found high correlations between Hb content and key morphometric parameters such as area (0.87 and 0.87) and width (0.86 and 0.87) for the ABS and RCS ROIs, respectively (Table). Conversely, the "mean grey value" parameter exhibited negligible correlation (ABS 0.02 and RCS 0.23), likely due to the two-dimensional constraints of the imaging process, interference from sclerotised abdominal structures, and the limited ability of grayscale intensity to reflect internal volume.

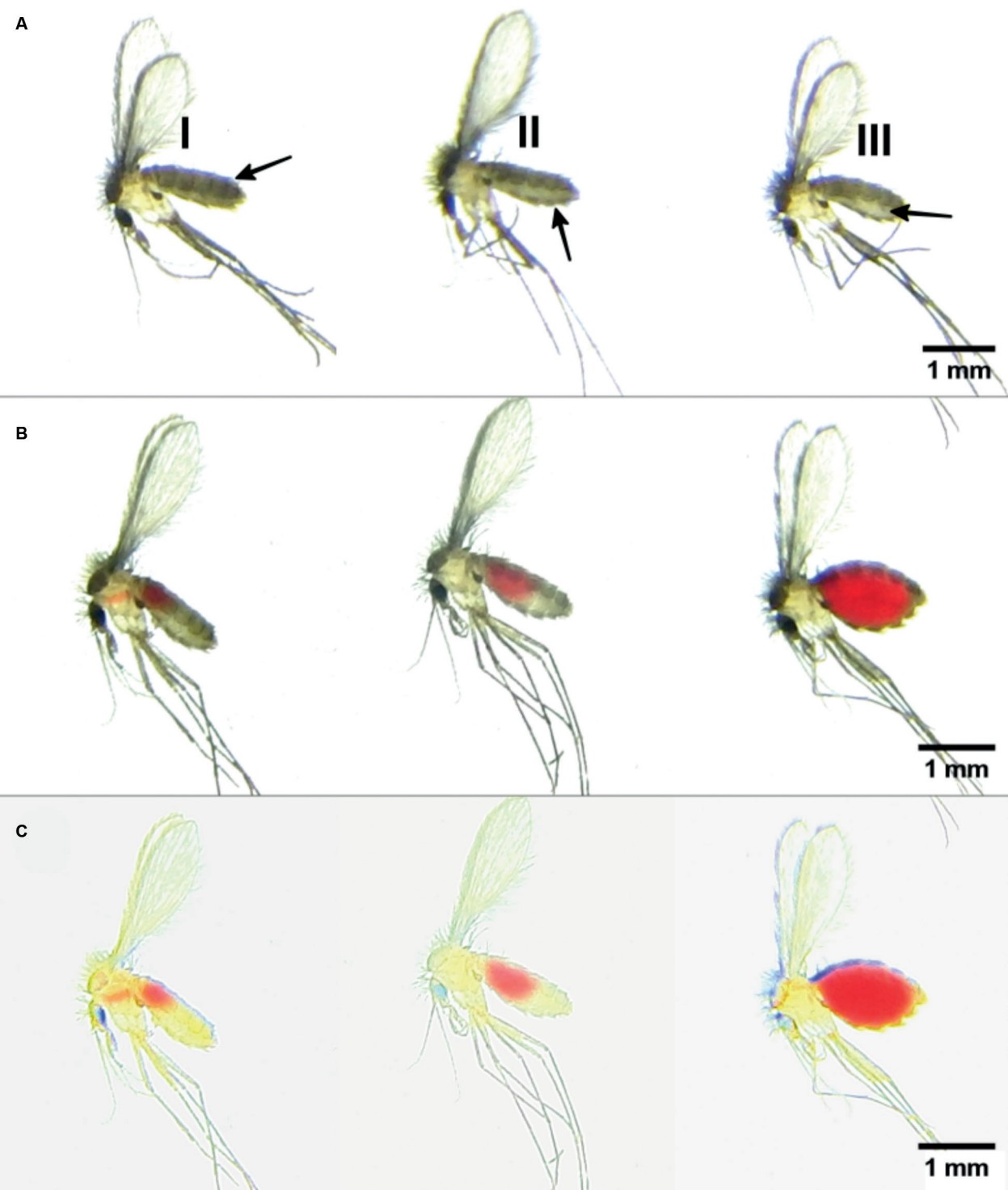

Fig. 2: original red, green and blue (RGB) images of three unfed *Lutzomyia longipalpis* females (control group, A) and three females that fed on hamsters (experimental group, B-C). The same blood-fed females after pseudo-flat field correction (BioVoxxel), highlighting red coloration in the digestive tract. Arrows indicate tergal plates (I), sternal plates (II), and pleural membrane (III).

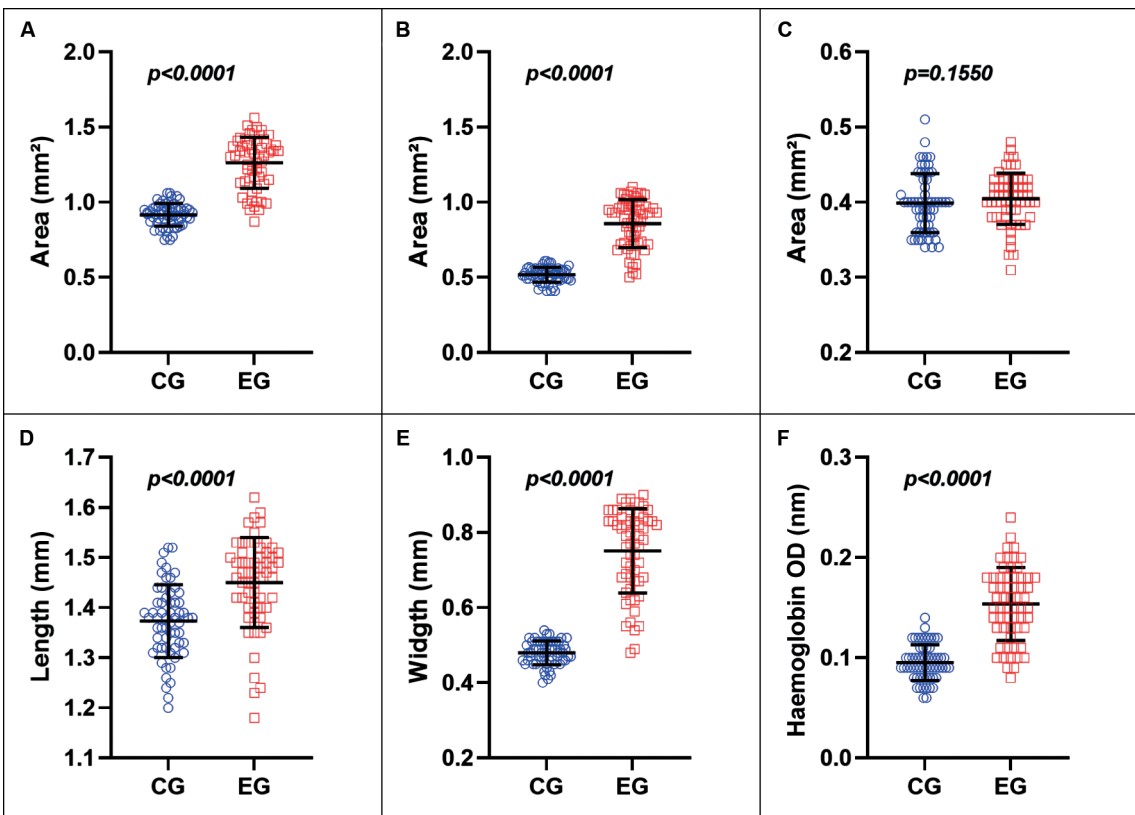

Fig. 3: comparison of image body regions and haemoglobin (Hb) optical density (OD) (540 nm) parameters between *Lutzomyia longipalpis* females in the control group (CG, blue) and experimental group (EG, red). (A-B) Total body area (TBS) and abdominal area (ABS) (mm²); (C) TBS minus ABS area (mm²); (D-E) abdominal length (mm) and width (mm); (F) Hb OD (540 nm). Mann-Whitney statistical tests.

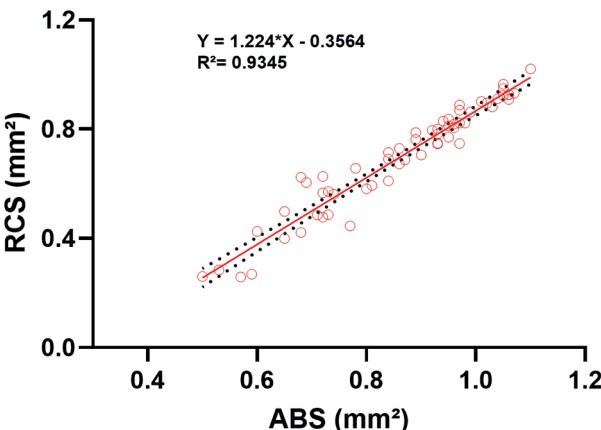

Fig. 4: linear regression analysis was used to evaluate blood feeding in 59 females *Lutzomyia longipalpis* [experimental group (EG)] comparing the areas (mm²) of the red abdominal selection (RCS) with the abdominal region (ABS) obtained from image processing. Dashed lines indicate confidence interval.

While Hb quantification remains a sensitive benchmark for measuring blood intake, it is inherently destructive, chemically hazardous, and unsuitable for longitudinal studies. In contrast, the image-based technique described here is non-invasive, reusable, and cost-effective, offering a viable, especially powerful alternative when paired with open-source tools like Fiji/ImageJ.

The gravimetric method, though widely used, also presents important constraints, particularly the inability to resolve individual blood intake levels in pooled or subtly engorged specimens without high-precision analytical balances. Our image-based approach effectively overcomes these limitations and introduces a practical solution for fine-scale phenotyping in vector biology studies.

Theoretically, this method is compatible with tracking live insects. This is important because many aspects of sand fly biology remain poorly understood - largely because, of the 927 known species, only 21 have established laboratory colonies.[40,41] The development of non-destructive methodologies suitable for repeated measurements provides new opportunities to clarify various aspects of sand fly physiology, particularly those triggered by blood ingestion. Although studies have shown that sand flies anaesthetised with cold or $CO_2$[42,43,44,45] can be used to initiate new colonies or to oviposit, the present study did not evaluate the recovery of individuals following cold anaesthesia at -20ºC.

In conclusion, our results demonstrate that image-based analysis constitutes a powerful, accessible, and scientifically rigorous tool for the indirect quantification of blood meals in *Lu. longipalpis*.

We support the use of image and Hb-based approaches to enhance precision, reproducibility, and scope across diverse experimental frameworks for estimating blood meals in vector research.

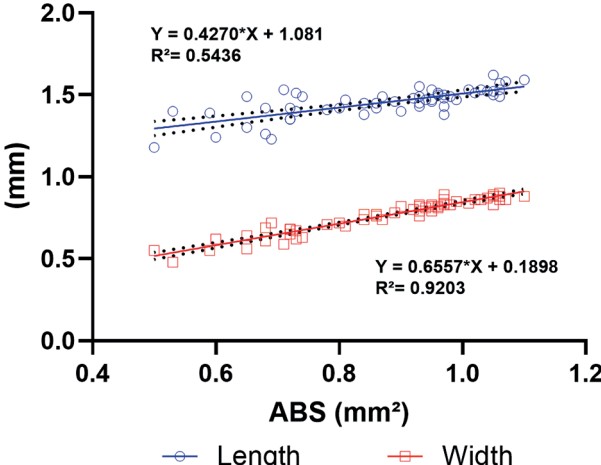

Fig. 5: linear regression models relating width (red, mm) and length (blue, mm) of the abdominal region (ABS) to total area (mm²), obtained through image analysis of blood-fed of 59 female *Lutzomyia longipalpis* fed with blood [experimental group (EG)]. Abdominal width and length were estimated from the major (~length) and minor (~width) axes calculated by the Fit Ellipse tool. Dashed lines indicate confidence interval.

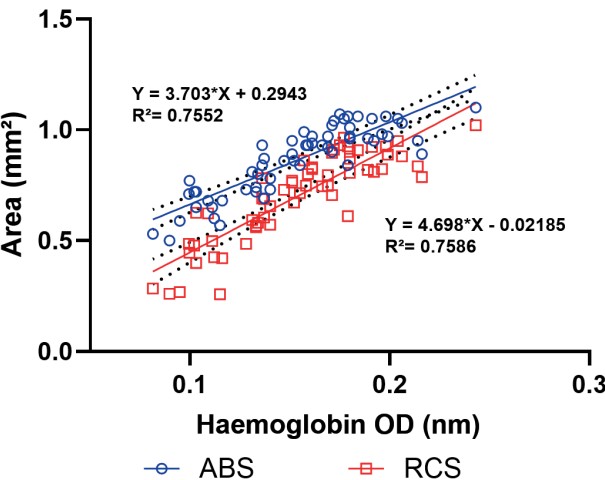

Fig. 6: linear regression models for the area parameters (mm²) of the abdominal region (ABS) and the red-coloured abdominal selection (RCS) as functions of optical density (OD) (540 nm) values obtained from the haemoglobin (Hb) assay in 59 females *Lutzomyia longipalpis* [experimental group (EG)]. Dashed lines indicate confidence interval (CI).

## TABLE

Pearson correlation coefficients between optical density(OD) (540 nm) values from the haemoglobin (Hb) assay and various image-derived parameters of abdominal selection (ABS) and red-coloured selection (RCS) of blood-fed *Lutzomyia longipalpis* females

|  | ROI | |
| --- | --- | --- |
| Parameter | ABS (n = 59) | RCS (n = 59) |
| Area | 0.87 | 0.87 |
| Mean | 0.02 | 0.23 |
| Length (Major) | 0.55 | 0.79 |
| Width (Minor) | 0.86 | 0.87 |
| Circularity | 0.66 | 0.15 |
| IntDen | 0.70 | 0.81 |
| Round | 0.72 | 0.33 |

ROI: region of interest.

## ACKNOWLEDGEMENTS

To César Nonato de Oliveira for technical assistance.

## AUTHORS' CONTRIBUTION

LMC - conception of the idea and design of the study, conducting experiments, data analysis and interpretation, draft writing; MRVS - draft writing, obtaining funding; LAG and GPC - conducting experiments; GCDP and RNA - critical review; NFG - critical review and obtaining funding; MHP - conception of the idea and design of the study, data analysis and interpretation, draft writing, obtaining funding and project coordination. The authors declare that they have no conflict of interest.

## DATA AVAILABILITY

All relevant data have been included in the manuscript.

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

# OPEN PEER REVIEW

Memórias do IOC thanks the anonymous reviewers for their contribution to the peer review of this work.

## FIRST REVIEW ROUND

REVIEWERS' COMMENTS

### REVIEWER #1

The manuscript from Costa et al. presents some correlations between body image parameters and amount of blood ingested in laboratory-reared Lutzomyia longipalpis. It is an interesting approach and well-presented report. However, the authors highly overstate the significance of their findings. There was no evidence that this technique may be used for live insects, or to follow sand fly behavior or any other parameter related to vector competence after the blood feed. Besides that, it is not clear if this technique will be valid in the future for other sand fly species of to field insects. In this respect, the Abstract, Introduction and Discussions must be thoroughly revised.

### REVIEWER #2

The manuscript addresses an important methodological challenge in sand fly research by proposing an image-based approach to estimate blood meal volume in Lutzomyia longipalpis, validated against hemoglobinometry. The study is well contextualized, the introduction highlights the biological and epidemiological relevance of blood meal quantification, and the results demonstrate strong correlations between abdominal morphometrics (particularly width and RCS area) and hemoglobin OD. The approach is low-cost, accessible, and has the potential to advance experimental studies on sand fly physiology and vector competence. These are clear strengths of your work.

That said, I believe the manuscript would benefit from several revisions before it can be considered for publication:

1- Non-destructive claim: The method is presented as "non-destructive," but all insects were euthanized prior to image analysis. Unless viability (survival, oviposition) is demonstrated, I suggest moderating this claim throughout the manuscript or, alternatively, adding a small subset of live females to show in vivo imaging feasibility.

2- Hemoglobin calibration: The standard curve was prepared with human blood, while the sand flies fed on hamster blood. Hematocrit and hemoglobin concentration vary between host species, which may affect the conversion of OD values to absolute volumes. Please clarify this point and either justify the choice of human blood or acknowledge this limitation in the Discussion.

3- Image acquisition and analysis: For reproducibility, more details are needed on image acquisition (fixed exposure/illumination, camera-sample distance) and analysis (parameters used for pseudo-flat field correction, thresholding method). Scale bars should be included in all figures. Figure 1 is a good start, but it would help to provide additional screenshots of the segmentation pipeline or a Fiji macro as supplementary material.

4- Statistical analysis: Since females were fed in groups on the same hamster, there is a potential clustering effect that simple tests (t-test, Mann-Whitney) and linear regressions do not account for. I recommend reanalyzing key variables with mixed-effects models (batch/hamster ID as random effect) and reporting effect sizes with 95% confidence intervals, in addition to p-values.

5- Sample exclusions: Four engorged females were excluded due to low OD values. Exclusion criteria should not be subjective; please define clear a priori thresholds, report final sample sizes for each group at each stage, and consider repeating technical replicates on a subset to assess reproducibility.

6- Figures and Table: Figures should include scale bars, error bars or confidence intervals, and indicate the statistical test and p-values directly. In Table 1, please use decimal points consistently and report N, p-values, and confidence intervals for the coefficients. In Figure 6, the regression equations and $R^2$ values are presented, which is excellent; I recommend also adding 95% confidence intervals to the regression lines.

7- Discussion: The practical advantages of the method are well argued, but I suggest moderating claims about reusing individuals until viability data are shown. Please also discuss briefly why "mean gray value" performed poorly (e.g., cuticle sclerotization, 2D limitations, sensitivity to illumination) and the implications for standardizing image acquisition. Finally, acknowledge that analyses were restricted to a single colony/lineage and to a single post-feeding time point (immediately after feeding). It would strengthen the manuscript to outline how the method might be generalized across digestion time windows (6-48 h) and across other Lu. longipalpis populations or related species.

8- Animal anesthesia: The description is incomplete. The dose is reported only as volume per weight (100 μL/100 g), without drug concentration (mg/kg), route of administration, formulation, or manufacturer. For reproducibility and compliance with animal welfare standards, please provide these details and indicate whether analgesia was used or considered.

In summary, this is a promising and valuable study with strong potential for methodological impact.

**REVIEWER #3**

The manuscript presents a method described as non-invasive and innovative, aiming to address the limitations of traditional, destructive techniques for estimating blood meal size in sand flies. While the non-destructive aspect is relevant, especially for studies involving live insect monitoring, the proposed image-based approach lacks true novelty, since Volvofa et al. (2024) published an article using measurements that can be performed without images, with equal credibility.

**AUTHORS' RESPONSE TO THE REVIEWERS**

Dear Editor,
We have carefully considered the reviewer's comments and revised the manuscript in accordance with their suggestions. We would also like, if possible, to include Reviewer 2 in the acknowledgements.

Yours sincerely,
Marcos Horácio Pereira

REVIEWER COMMENTS:
Reviewer: 1
The manuscript from Costa et al. presents some correlations between body image parameters and amount of blood ingested in laboratory-reared Lutzomyia longipalpis. It is an interesting approach and well presented report.

However, the authors highly overstate the significance of their findings. There was no evidence that this technique may be used for live insects, or to follow sand fly behavior or any other parameter related to vector competence after the blood feed. Besides that, it is not clear if this technique will be valid in the future for other sand fly species of to field insects. In this respect, the Abstract, Introduction and Discussions must be thoroughly revised.

Response:
We thank the reviewer for their thoughtful comments. In the present study, all female sand flies were cold-anaesthetised at -20°C for 10 minutes, photographed, and subsequently sacrificed for haemoglobin quantification, as our aim was to compare image-derived parameters (e.g., area) with absorbance at 540 nm. We did not assess insect recovery following cold anaesthesia at –20 °C. Accordingly, we have revised specific sections of the manuscript (Abstract, Introduction, and Discussion – see below) to avoid giving the impression that any parameter other than image capture and haemoglobin quantification was evaluated.

As our results showed strong concordance with haemoglobin measurements - considered the gold-standard method for estimating blood meal size - we believe that image-based analysis may also be applicable for estimating blood meal size in other sand fly species; however, this should be tested experimentally. Because the primary equipment needed for image capture is a digital camera, this methodology can, with minor adjustments, be adapted for field use.

i) The abstract section below has been rewritten:
See lines 37 - 42 in the manuscript reviewed.
ii) The final sentence of the Introduction has been removed:
See lines 108 - 110 in the manuscript reviewed.
iii) The paragraph below in the Discussion has been rewritten:
See lines 337 - 355 in the manuscript reviewed.
iv) The following sentence in the Discussion has been removed:
See lines 358 - 360 in the manuscript reviewed.

Reviewer: 2
The manuscript addresses an important methodological challenge in sand fly research by proposing an image-based approach to estimate blood meal volume in Lutzomyia longipalpis, validated against hemoglobinometry. The study is well contextualized, the introduction highlights the biological and epidemiological relevance of blood meal quantification, and the results demonstrate strong correlations between abdominal morphometrics (particularly width and RCS area) and hemoglobin OD. The approach is low-cost, accessible, and has the potential to advance experimental studies on sand fly physiology and vector competence. These are clear strengths of your work. That said, I believe the manuscript would benefit from several revisions before it can be considered for publication:

1- Non-destructive claim: The method is presented as "non-destructive," but all insects were euthanized prior to image analysis. Unless viability (survival, oviposition) is demonstrated, I suggest moderating this claim throughout the manuscript or, alternatively, adding a small subset of live females to show in vivo imaging feasibility.

Response:
We thank you for your comments. In this study, all female specimens were cold-anaesthetised at –20 °C for 10 minutes, photographed, and subsequently sacrificed for haemoglobin quantification, as our aim was to compare

image-derived parameters (e.g., area) with absorbance at 540 nm. We did not assess insect recovery following cold anaesthesia at –20 °C. Accordingly, we have revised sections of the manuscript (Abstract, Introduction, and Discussion) to avoid giving the impression that any parameter other than image acquisition and haemoglobin quantification was evaluated.

i) The abstract section below has been rewritten:

See lines 37 - 42 in the manuscript reviewed.

ii) The final sentence of the Introduction has been removed:

See lines 108 - 110 in the manuscript reviewed.

iii) The paragraph below in the Discussion has been rewritten:

See lines 337 - 335 in the manuscript reviewed.

iv) The following sentence in the Discussion has been removed:

See lines 358 - 360 in the manuscript reviewed.

2- Hemoglobin calibration: The standard curve was prepared with human blood, while the sand flies fed on hamster blood. Hematocrit and hemoglobin concentration vary between host species, which may affect the conversion of OD values to absolute volumes. Please clarify this point and either justify the choice of human blood or acknowledge this limitation in the Discussion.

Response:

Thank you for the comment. We agree that using hamster blood rather than human blood would have been preferable for constructing the standard curve, although some studies have used human haemoglobin for this purpose (Sant'Anna et al., 2010; Pruzinova et al., 2015). In humans, the haematocrit ranges from 37–52% and haemoglobin concentration from 12.0–18.0 g/dl, whereas hamsters present a haematocrit of $42.0 \pm 1.9\%$ and a haemoglobin concentration of $15.2 \pm 0.6$ g/dl (Lewis, 1996 – Comparative Hemostasis in Vertebrates, Plenum Press, New York).

However, considering that most mammalian haemoglobins are expected to yield similar absorbance values, although using a species-matched standard is best practice for research and clinical accuracy, and given the similarity in haematocrit and haemoglobin concentration between hamsters and humans, no major distortion in our results is expected.

3- Image acquisition and analysis: For reproducibility, more details are needed on image acquisition (fixed exposure/illumination, camera-sample distance) and analysis (parameters used for pseudo-flat field correction, thresholding method). Scale bars should be included in all figures. Figure 1 is a good start, but it would help to provide additional screenshots of the segmentation pipeline or a Fiji macro as supplementary material.

Response:

Thank you for your comments and suggestions. In the "Image Acquisition and Analysis" section of the methodology, we have included additional details on the image-acquisition process as well as on the use of the pseudo–flat field correction tool. In the "Selection of Regions of Interest (ROIs)" section, we have added information regarding the thresholding and colour-thresholding functions. Scale bars in Figure 2 have been added and/or enhanced. We have also prepared a supplementary figure containing screenshots illustrating the steps involved in obtaining the Regions of Interest (ROIs) shown in Figure 1 (Supplementary figure 1)

i) In the "Image Acquisition and Analysis" section, we have rewritten and incorporated the following information into the text:

See lines 147 - 153 in the manuscript reviewed.

And

See lines 157 - 162 in the manuscript reviewed.

ii) In the "Selection of Regions of Interest (ROIs)" section of the methodology, information regarding the thresholding and colour-thresholding tools was added:

See lines 167 - 171 in the manuscript reviewed.

4- Statistical analysis: Since females were fed in groups on the same hamster, there is a potential clustering effect that simple tests (t-test, Mann-Whitney) and linear regressions do not account for. I recommend reanalyzing key variables with mixed-effects models (batch/hamster ID as random effect) and reporting effect sizes with 95% confidence intervals, in addition to p-values.

Response:

All 60 females in the experimental group fed on the same hamster; therefore, assessment of a potential clustering effect does not apply. We have also revised the text in this section of the methodology (see below).

i) In the methodological sentence below, the word hamsters had inadvertently been written in the plural, and was corrected to hamster (which may have caused the confusion):

See line 139 in the manuscript reviewed.

5- Sample exclusions: Four engorged females were excluded due to low OD values. Exclusion criteria should not be subjective; please define clear a priori thresholds, report final sample sizes for each group at each stage, and consider repeating technical replicates on a subset to assess reproducibility.

Response:

We thank you for your comment. We agree that criteria for outlier exclusion must be clearly defined. Therefore, we have added the exclusion criteria used in our analyses to the "Statistical Analysis" section of the methodology. Outliers were excluded when, after fitting a linear regression, (1) absolute studentised residuals were > 3 and (2) Cook's distances were > 4/n (see Altman & Krzywinski, 2016). Further details are provided below.

Using this criterion, only one (insect no.52) of the 60 insects analysed was excluded from the comparison between image-based measurements (ABS and RCS areas) and haemoglobin quantification (OD at 540 nm) (see PDF Reviewer 2 - Response 5; figs 1, 2 and 3).

When this specimen is removed, the linear regression coefficients increase from $R^2 = 0.699$ to $0.759$ (OD × ABS area) and from $R^2 = 0.666$ to $0.759$ (OD × RCS area) (see PDF Reviewer 2 - Response 5; figs 4 and 5).

Accordingly, we included all 59 remaining insects in the comparison between image-derived measurements (ABS and RCS areas) and haemoglobin quantification (OD at 540 nm), recalculated all linear regression plots (Figures 4, 5 and 6), and updated the correlation shown in Table I. Below, we include the outlier exclusion criteria added to the "Statistical Analysis" section. We also corrected the corresponding information (excluded points) in the Results section, under item 3.3 ("Analysis of Blood-Fed Females Based on Abdominal Colour").

i) We have included the following information on outlier exclusion in the "Statistical Analysis" section of the methodology:

See line 210 - 212 in the manuscript reviewed.

ii) We also corrected this information (excluded points) in the wording of item 3.3 ("Analysis of Blood-Fed Females Based on Abdominal Colour") in the Results section:

See line 261 - 269 in the manuscript reviewed.

6- Figures and Table: Figures should include scale bars, error bars or confidence intervals, and indicate the statistical test and p-values directly. In Table 1, please use decimal points consistently and report N, p-values, and confidence intervals for the coefficients. In Figure 6, the regression equations and $R^2$ values are presented, which is excellent; I recommend also adding 95% confidence intervals to the regression lines.

Response:

Thank you for your comments and suggestions. All figures and the table have been revised to correct formatting (replacing commas with full stops), standardize the number of decimal places (Table I), highlight scale bars (Figure 2), include 95% confidence intervals (Figures 4, 5, and 6), and replace asterisks with the corresponding p-values (Figure 2). The figure legends now specify the statistical test used and/or the sample size (N) for each parameter analysed.

7- Discussion: The practical advantages of the method are well argued, but I suggest moderating claims about reusing individuals until viability data are shown. Please also discuss briefly why "mean gray value" performed poorly (e.g., cuticle sclerotization, 2D limitations, sensitivity to illumination) and the implications for standardizing image acquisition. Finally, acknowledge that analyses were restricted to a single colony/lineage and to a single post-feeding time point (immediately after feeding). It would strengthen the manuscript to outline how the method might be generalized across digestion time windows (6-48 h) and across other Lu. longipalpis populations or related species.

Response:

Thank you for your response. We agree with your comment; please refer to our reply to the first question. Colour indeed helped distinguish blood-fed from unfed females ("distinct reddish abdominal coloration was observed in engorged females, contrasting with the brownish abdomen of unfed specimens"). The mean gray value of fed (93.4 ± 11.00) and unfed (142.1 ± 10.36) females, together with the corresponding statistical analysis, were provided in the results. We corrected Table 1 by adding the correlation between the mean gray values (ABS and RCS) and optical density at 540 nm, and the discussion of this part was rewritten.

As our results showed strong agreement with haemoglobin quantification—the gold-standard method for estimating blood meal size—we believe that image analysis may also be applied to estimate blood meal size in other sand fly species. Image analysis may also be useful for characterising physiological processes, such as changes in the colour and/or volume of ingested blood (Dolmatova, 1942) during digestion (6–48 h). However, the application of image analysis to estimate blood meal size in other sand fly species, or to characterise blood digestion itself, must be empirically tested.

i) The information of the mean gray value of fed and unfed females was added to the results (Blood Feeding-Induced Changes in Body Size):

See line 251 - 252 in the manuscript reviewed.

ii) The following paragraph was added to the Discussion:

See line 321 - 324 in the manuscript reviewed.

8- Animal anesthesia: The description is incomplete. The dose is reported only as volume per weight (100 μL/100 g), without drug concentration (mg/kg), route of administration, formulation, or manufacturer. For reproducibility and compliance with animal welfare standards, please provide these details and indicate whether analgesia was used or considered. In summary, this is a promising and valuable study with strong potential for methodological impact.

Response:

We thank you for your observations and have amended the text, accordingly, as detailed below.

i) The requested information has been included in the "Sand Fly Rearing" section of the methodology:

See line 123 - 130 in the manuscript reviewed.

Reviewer: 3

The manuscript presents a method described as non-invasive and innovative, aiming to address the limitations of traditional, destructive techniques for estimating blood meal size in sand flies. While the non-destructive aspect is relevant, especially for studies involving live insect monitoring, the proposed image-based approach lacks true novelty, since Volvofa et al. (2024) published an article using measurements that can be performed without images, with equal credibility.

Response:

Thank you for your comments. Estimating the blood meal size of sand flies is not straightforward, regardless of the method employed, and requires both expertise and methodological rigour on the part of the researcher to obtain reliable results. Given that our findings showed strong agreement with haemoglobin quantification (the gold-standard method for estimating blood meal size) we believe that image analysis expands the methodological options available to researchers working with L. longipalpis, with the potential to be applied to other sand fly species as well. Among the cited studies is the publication by Volvofa et al. (2024), which provides important contributions to the field.

## SECOND REVIEW ROUND

### REVIEWERS' COMMENTS

### REVIEWER #1

No comments.

### REVIEWER #2

Dear authors,

I would like to congratulate you on the revised version of your manuscript. The improvements made are substantial and clearly reflect a careful and thoughtful response to the reviewers' comments. The manuscript is now much clearer, more coherent, and methodologically transparent, and it convincingly demonstrates the value of your image-based approach for estimating blood meal size in sand flies.

I particularly commend you for moderating the language regarding the non-destructive nature of the methodology and its applicability to live insects. This more balanced framing strengthens the scientific integrity of the work and aligns well with the experimental validation presented. As revised, the study offers a relevant and impactful contribution to vector biology and experimental entomology.

I would like to offer one minor suggestion, purely from a conceptual and editorial perspective. Given the current, more nuanced discussion, you might consider whether a title emphasizing the approach as an "image-based proxy", rather than explicitly "non-destructive", could better reflect the scope of the validation performed and avoid any potential perception of extrapolation. This is not intended as a criticism and should not be seen as a barrier to publication, but rather as an optional refinement to maintain full coherence between title and text.

Aside from this point, I see no remaining issues that would prevent publication. I congratulate you once again on a very solid and well-executed piece of work.

(The name of the corresponding author, Marcos Horácio Pereira, does not currently appear in the author list.)

### REVIEWER #3

No comments.

