## [Reviewer Report · FIRST REVIEW ROUND - REVIEWERS COMMENTS]

## REVIEWER #1

The manuscript from Costa et al. presents some correlations between body image parameters and amount of blood ingested in laboratory-reared *Lutzomyia longipalpis*.

It is an interesting approach and well-presented report. However, the authors highly overstate the significance of their findings.

There was no evidence that this technique may be used for live insects, or to follow sand fly behavior or any other parameter related to vector competence after the blood feed.

Besides that, it is not clear if this technique will be valid in the future for other sand fly species of to field insects.

In this respect, the Abstract, Introduction and Discussions must be thoroughly revised.

## REVIEWER #2

The manuscript addresses an important methodological challenge in sand fly research by proposing an image-based approach to estimate blood meal volume in *Lutzomyia longipalpis*, validated against hemoglobinometry.

The study is well contextualized, the introduction highlights the biological and epidemiological relevance of blood meal quantification, and the results demonstrate strong correlations between abdominal morphometrics (particularly width and RCS area) and hemoglobin OD.

The approach is low-cost, accessible, and has the potential to advance experimental studies on sand fly physiology and vector competence.

These are clear strengths of your work.

That said, I believe the manuscript would benefit from several revisions before it can be considered for publication:

1- Non-destructive claim: The method is presented as “non-destructive,” but all insects were euthanized prior to image analysis.

Unless viability (survival, oviposition) is demonstrated, I suggest moderating this claim throughout the manuscript or, alternatively, adding a small subset of live females to show in vivo imaging feasibility.

2- Hemoglobin calibration: The standard curve was prepared with human blood, while the sand flies fed on hamster blood.

Hematocrit and hemoglobin concentration vary between host species, which may affect the conversion of OD values to absolute volumes.

Please clarify this point and either justify the choice of human blood or acknowledge this limitation in the Discussion.

3- Image acquisition and analysis: For reproducibility, more details are needed on image acquisition (fixed exposure/illumination, camera-sample distance) and analysis (parameters used for pseudo-flat field correction, thresholding method).

Scale bars should be included in all figures. Figure 1 is a good start, but it would help to provide additional screenshots of the segmentation pipeline or a Fiji macro as supplementary material.

4- Statistical analysis: Since females were fed in groups on the same hamster, there is a potential clustering effect that simple tests (t-test, Mann-Whitney) and linear regressions do not account for.

I recommend reanalyzing key variables with mixed-effects models (batch/hamster ID as random effect) and reporting effect sizes with 95% confidence intervals, in addition to p-values.

5- Sample exclusions: Four engorged females were excluded due to low OD values. Exclusion criteria should not be subjective;

please define clear a priori thresholds, report final sample sizes for each group at each stage, and consider repeating technical replicates on a subset to assess reproducibility.

6- Figures and Table: Figures should include scale bars, error bars or confidence intervals, and indicate the statistical test and p-values directly.

In Table 1, please use decimal points consistently and report N, p-values, and confidence intervals for the coefficients.

In Figure 6, the regression equations and R² values are presented, which is excellent;

I recommend also adding 95% confidence intervals to the regression lines.

7- Discussion: The practical advantages of the method are well argued, but I suggest moderating claims about reusing individuals until viability data are shown.

Please also discuss briefly why “mean gray value” performed poorly (e.g., cuticle sclerotization, 2D limitations, sensitivity to illumination) and the implications for standardizing image acquisition.

Finally, acknowledge that analyses were restricted to a single colony/lineage and to a single post-feeding time point (immediately after feeding).

It would strengthen the manuscript to outline how the method might be generalized across digestion time windows (6-48 h) and across other *Lu. longipalpis* populations or related species.

8- Animal anesthesia: The description is incomplete. The dose is reported only as volume per weight (100 µL/100 g), without drug concentration (mg/kg), route of administration, formulation, or manufacturer.

For reproducibility and compliance with animal welfare standards, please provide these details and indicate whether analgesia was used or considered.

In summary, this is a promising and valuable study with strong potential for methodological impact.

## REVIEWER #3

The manuscript presents a method described as non-invasive and innovative, aiming to address the limitations of traditional, destructive techniques for estimating blood meal size in sand flies.

While the non-destructive aspect is relevant, especially for studies involving live insect monitoring, the proposed image-based approach lacks true novelty, since Volvofa et al.

(2024) published an article using measurements that can be performed without images, with equal credibility.

## AUTHORS’ RESPONSE TO THE REVIEWERS

Dear Editor,

We have carefully considered the reviewer’s comments and revised the manuscript in accordance with their suggestions.

We would also like, if possible, to include Reviewer 2 in the acknowledgements.

Yours sincerely,

Marcos Horácio Pereira

REVIEWER COMMENTS:

Reviewer: 1

The manuscript from Costa et al. presents some correlations between body image parameters and amount of blood ingested in laboratory-reared *Lutzomyia longipalpis*.

It is an interesting approach and well presented report.

However, the authors highly overstate the significance of their findings.

There was no evidence that this technique may be used for live insects, or to follow sand fly behavior or any other parameter related to vector competence after the blood feed.

Besides that, it is not clear if this technique will be valid in the future for other sand fly species of to field insects.

In this respect, the Abstract, Introduction and Discussions must be thoroughly revised.

Response:

We thank the reviewer for their thoughtful comments. In the present study, all female sand flies were cold-anaesthetised at -20°C for 10 minutes, photographed, and subsequently sacrificed for haemoglobin quantification, as our aim was to compare image-derived parameters (e.g., area) with absorbance at 540 nm.

We did not assess insect recovery following cold anaesthesia at –20 °C.

Accordingly, we have revised specific sections of the manuscript (Abstract, Introduction, and Discussion – see below) to avoid giving the impression that any parameter other than image capture and haemoglobin quantification was evaluated.

As our results showed strong concordance with haemoglobin measurements - considered the gold-standard method for estimating blood meal size - we believe that image-based analysis may also be applicable for estimating blood meal size in other sand fly species;

however, this should be tested experimentally. Because the primary equipment needed for image capture is a digital camera, this methodology can, with minor adjustments, be adapted for field use.

i) The abstract section below has been rewritten:

See lines 37 - 42 in the manuscript reviewed.

ii) The final sentence of the Introduction has been removed:

See lines 108 - 110 in the manuscript reviewed.

iii) The paragraph below in the Discussion has been rewritten:

See lines 337 - 355 in the manuscript reviewed.

iv) The following sentence in the Discussion has been removed:

See lines 358 - 360 in the manuscript reviewed.

Reviewer: 2

The manuscript addresses an important methodological challenge in sand fly research by proposing an image-based approach to estimate blood meal volume in *Lutzomyia longipalpis*, validated against hemoglobinometry.

The study is well contextualized, the introduction highlights the biological and epidemiological relevance of blood meal quantification, and the results demonstrate strong correlations between abdominal morphometrics (particularly width and RCS area) and hemoglobin OD.

The approach is low-cost, accessible, and has the potential to advance experimental studies on sand fly physiology and vector competence.

These are clear strengths of your work. That said, I believe the manuscript would benefit from several revisions before it can be considered for publication:

1- Non-destructive claim: The method is presented as “non-destructive,” but all insects were euthanized prior to image analysis.

Unless viability (survival, oviposition) is demonstrated, I suggest moderating this claim throughout the manuscript or, alternatively, adding a small subset of live females to show in vivo imaging feasibility.

Response:

We thank you for your comments. In this study, all female specimens were cold-anaesthetised at –20 °C for 10 minutes, photographed, and subsequently sacrificed for haemoglobin quantification, as our aim was to compare image-derived parameters (e.g., area) with absorbance at 540 nm.

We did not assess insect recovery following cold anaesthesia at –20 °C.

Accordingly, we have revised sections of the manuscript (Abstract, Introduction, and Discussion) to avoid giving the impression that any parameter other than image acquisition and haemoglobin quantification was evaluated.

i) The abstract section below has been rewritten:

See lines 37 - 42 in the manuscript reviewed.

ii) The final sentence of the Introduction has been removed:

See lines 108 - 110 in the manuscript reviewed.

iii) The paragraph below in the Discussion has been rewritten:

See lines 337 - 335 in the manuscript reviewed.

iv) The following sentence in the Discussion has been removed:

See lines 358 - 360 in the manuscript reviewed.

2- Hemoglobin calibration: The standard curve was prepared with human blood, while the sand flies fed on hamster blood.

Hematocrit and hemoglobin concentration vary between host species, which may affect the conversion of OD values to absolute volumes.

Please clarify this point and either justify the choice of human blood or acknowledge this limitation in the Discussion.

Response:

Thank you for the comment. We agree that using hamster blood rather than human blood would have been preferable for constructing the standard curve, although some studies have used human haemoglobin for this purpose (Sant’Anna et al., 2010; Pruzinova et al., 2015).

In humans, the haematocrit ranges from 37–52% and haemoglobin concentration from 12.0–18.0 g/dl, whereas hamsters present a haematocrit of 42.0 ± 1.9% and a haemoglobin concentration of 15.2 ± 0.6 g/dl (Lewis, 1996 – Comparative Hemostasis in Vertebrates, Plenum Press, New York).

However, considering that most mammalian haemoglobins are expected to yield similar absorbance values, although using a species-matched standard is best practice for research and clinical accuracy, and given the similarity in haematocrit and haemoglobin concentration between hamsters and humans, no major distortion in our results is expected.

3- Image acquisition and analysis: For reproducibility, more details are needed on image acquisition (fixed exposure/illumination, camera-sample distance) and analysis (parameters used for pseudo-flat field correction, thresholding method).

Scale bars should be included in all figures. Figure 1 is a good start, but it would help to provide additional screenshots of the segmentation pipeline or a Fiji macro as supplementary material.

Response:

Thank you for your comments and suggestions. In the “Image Acquisition and Analysis” section of the methodology, we have included additional details on the image-acquisition process as well as on the use of the pseudo–flat field correction tool.

In the “Selection of Regions of Interest (ROIs)” section, we have added information regarding the thresholding and colour-thresholding functions.

Scale bars in Figure 2 have been added and/or enhanced.

We have also prepared a supplementary figure containing screenshots illustrating the steps involved in obtaining the Regions of Interest (ROIs) shown in Figure 1 (Supplementary figure 1)

i) In the “Image Acquisition and Analysis” section, we have rewritten and incorporated the following information into the text:

See lines 147 - 153 in the manuscript reviewed.

And

See lines 157 - 162 in the manuscript reviewed.

ii) In the “Selection of Regions of Interest (ROIs)” section of the methodology, information regarding the thresholding and colour-thresholding tools was added:

See lines 167 - 171 in the manuscript reviewed.

4- Statistical analysis: Since females were fed in groups on the same hamster, there is a potential clustering effect that simple tests (t-test, Mann-Whitney) and linear regressions do not account for.

I recommend reanalyzing key variables with mixed-effects models (batch/hamster ID as random effect) and reporting effect sizes with 95% confidence intervals, in addition to p-values.

Response:

All 60 females in the experimental group fed on the same hamster;

therefore, assessment of a potential clustering effect does not apply.

We have also revised the text in this section of the methodology (see below).

i) In the methodological sentence below, the word hamsters had inadvertently been written in the plural, and was corrected to hamster (which may have caused the confusion):

See line 139 in the manuscript reviewed.

5- Sample exclusions: Four engorged females were excluded due to low OD values. Exclusion criteria should not be subjective;

please define clear a priori thresholds, report final sample sizes for each group at each stage, and consider repeating technical replicates on a subset to assess reproducibility.

Response:

We thank you for your comment. We agree that criteria for outlier exclusion must be clearly defined.

Therefore, we have added the exclusion criteria used in our analyses to the “Statistical Analysis” section of the methodology.

Outliers were excluded when, after fitting a linear regression, (1) absolute studentised residuals were > 3 and (2) Cook’s distances were > 4/n (see Altman & Krzywinski, 2016).

Further details are provided below.

Using this criterion, only one (insect no.52) of the 60 insects analysed was excluded from the comparison between image-based measurements (ABS and RCS areas) and haemoglobin quantification (OD at 540 nm) (see PDF Reviewer 2 - Response 5; figs 1, 2 and 3).

When this specimen is removed, the linear regression coefficients increase from R² = 0.699 to 0.759 (OD × ABS area) and from R² = 0.666 to 0.759 (OD × RCS area) (see PDF Reviewer 2 - Response 5; figs 4 and 5).

Accordingly, we included all 59 remaining insects in the comparison between image-derived measurements (ABS and RCS areas) and haemoglobin quantification (OD at 540 nm), recalculated all linear regression plots (Figures 4, 5 and 6), and updated the correlation shown in Table I. Below, we include the outlier exclusion criteria added to the “Statistical Analysis” section.

We also corrected the corresponding information (excluded points) in the Results section, under item 3.3 (“Analysis of Blood-Fed Females Based on Abdominal Colour”).

i) We have included the following information on outlier exclusion in the “Statistical Analysis” section of the methodology:

See line 210 - 212 in the manuscript reviewed.

ii) We also corrected this information (excluded points) in the wording of item 3.3 (“Analysis of Blood-Fed Females Based on Abdominal Colour”) in the Results section:

See line 261 - 269 in the manuscript reviewed.

6- Figures and Table: Figures should include scale bars, error bars or confidence intervals, and indicate the statistical test and p-values directly.

In Table 1, please use decimal points consistently and report N, p-values, and confidence intervals for the coefficients.

In Figure 6, the regression equations and R² values are presented, which is excellent;

I recommend also adding 95% confidence intervals to the regression lines.

Response:

Thank you for your comments and suggestions. All figures and the table have been revised to correct formatting (replacing commas with full stops), standardize the number of decimal places (Table I), highlight scale bars (Figure 2), include 95% confidence intervals (Figures 4, 5, and 6), and replace asterisks with the corresponding p-values (Figure 2).

The figure legends now specify the statistical test used and/or the sample size (N) for each parameter analysed.

7- Discussion: The practical advantages of the method are well argued, but I suggest moderating claims about reusing individuals until viability data are shown.

Please also discuss briefly why “mean gray value” performed poorly (e.g., cuticle sclerotization, 2D limitations, sensitivity to illumination) and the implications for standardizing image acquisition.

Finally, acknowledge that analyses were restricted to a single colony/lineage and to a single post-feeding time point (immediately after feeding).

It would strengthen the manuscript to outline how the method might be generalized across digestion time windows (6-48 h) and across other *Lu. longipalpis* populations or related species.

Response:

Thank you for your response. We agree with your comment; please refer to our reply to the first question.

Colour indeed helped distinguish blood-fed from unfed females (“distinct reddish abdominal coloration was observed in engorged females, contrasting with the brownish abdomen of unfed specimens”).

The mean gray value of fed (93.4 ± 11.00) and unfed (142.1 ± 10.36) females, together with the corresponding statistical analysis, were provided in the results.

We corrected Table 1 by adding the correlation between the mean gray values (ABS and RCS) and optical density at 540 nm, and the discussion of this part was rewritten.

As our results showed strong agreement with haemoglobin quantification—the gold-standard method for estimating blood meal size—we believe that image analysis may also be applied to estimate blood meal size in other sand fly species.

Image analysis may also be useful for characterising physiological processes, such as changes in the colour and/or volume of ingested blood (Dolmatova, 1942) during digestion (6–48 h).

However, the application of image analysis to estimate blood meal size in other sand fly species, or to characterise blood digestion itself, must be empirically tested.

i) The information of the mean gray value of fed and unfed females was added to the results (Blood Feeding-Induced Changes in Body Size):

See line 251 - 252 in the manuscript reviewed.

ii) The following paragraph was added to the Discussion:

See line 321 - 324 in the manuscript reviewed.

8- Animal anesthesia: The description is incomplete. The dose is reported only as volume per weight (100 µL/100 g), without drug concentration (mg/kg), route of administration, formulation, or manufacturer.

For reproducibility and compliance with animal welfare standards, please provide these details and indicate whether analgesia was used or considered.

In summary, this is a promising and valuable study with strong potential for methodological impact.

Response:

We thank you for your observations and have amended the text, accordingly, as detailed below.

i) The requested information has been included in the “Sand Fly Rearing” section of the methodology:

See line 123 - 130 in the manuscript reviewed.

Reviewer: 3

The manuscript presents a method described as non-invasive and innovative, aiming to address the limitations of traditional, destructive techniques for estimating blood meal size in sand flies.

While the non-destructive aspect is relevant, especially for studies involving live insect monitoring, the proposed image-based approach lacks true novelty, since Volvofa et al.

(2024) published an article using measurements that can be performed without images, with equal credibility.

Response:

Thank you for your comments. Estimating the blood meal size of sand flies is not straightforward, regardless of the method employed, and requires both expertise and methodological rigour on the part of the researcher to obtain reliable results.

Given that our findings showed strong agreement with haemoglobin quantification (the gold-standard method for estimating blood meal size) we believe that image analysis expands the methodological options available to researchers working with *L. longipalpis*, with the potential to be applied to other sand fly species as well.

Among the cited studies is the publication by Volvofa et al. (2024), which provides important contributions to the field.

---

## [Reviewer Report · REVIEWERS COMMENTS]

## REVIEWER #1

No comments.

## REVIEWER #2

Dear authors,

I would like to congratulate you on the revised version of your manuscript.

The improvements made are substantial and clearly reflect a careful and thoughtful response to the reviewers’ comments.

The manuscript is now much clearer, more coherent, and methodologically transparent, and it convincingly demonstrates the value of your image-based approach for estimating blood meal size in sand flies.

I particularly commend you for moderating the language regarding the non-destructive nature of the methodology and its applicability to live insects.

This more balanced framing strengthens the scientific integrity of the work and aligns well with the experimental validation presented.

As revised, the study offers a relevant and impactful contribution to vector biology and experimental entomology.

I would like to offer one minor suggestion, purely from a conceptual and editorial perspective.

Given the current, more nuanced discussion, you might consider whether a title emphasizing the approach as an “image-based proxy”, rather than explicitly “non-destructive”, could better reflect the scope of the validation performed and avoid any potential perception of extrapolation.

This is not intended as a criticism and should not be seen as a barrier to publication, but rather as an optional refinement to maintain full coherence between title and text.

Aside from this point, I see no remaining issues that would prevent publication.

I congratulate you once again on a very solid and well-executed piece of work.

(The name of the corresponding author, Marcos Horácio Pereira, does not currently appear in the author list.)

## REVIEWER #3

No comments.